# HINDSIGHT IS 20/20: LEVERAGING PAST TRAVERSALS TO AID 3D PERCEPTION

**Yurong You**[1]**, Katie Z Luo**[1]**, Xiangyu Chen**[1]**, Junan Chen**[1]**, Wei-Lun Chao**[2]**,**
**Wen Sun**[1]**, Bharath Hariharan**[1]**, Mark Campbell**[1]**,** and **Kilian Q. Weinberger**[1]
[1]Cornell University, Ithaca, NY      [2]The Ohio State University, Columbus, OH

## ABSTRACT

Self-driving cars must detect vehicles, pedestrians, and other traffic participants accurately to operate safely. Small, far-away, or highly occluded objects are particularly challenging because there is limited information in the LiDAR point clouds for detecting them. To address this challenge, we leverage valuable information from the past: in particular, data collected in past traversals of the same scene. We posit that these past data, which are typically discarded, provide rich contextual information for disambiguating the above-mentioned challenging cases. To this end, we propose a novel, end-to-end trainable HINDSIGHT framework to extract this contextual information from past traversals and store it in an easy-to-query data structure, which can then be leveraged to aid future 3D object detection of the same scene. We show that this framework is compatible with most modern 3D detection architectures and can substantially improve their average precision on multiple autonomous driving datasets, most notably by more than 300% on the challenging cases. Our code is available at https://github.com/YurongYou/Hindsight.

## 1 INTRODUCTION

To drive safely, a (semi-)autonomous vehicle needs to accurately detect and localize other participants, such as cars, buses, cyclists, or pedestrians who might walk onto the road at any time. Such an object detection task is an extremely challenging 3D perception problem, especially when it comes to small, highly occluded, or far-away objects. In spite of considerable progress on this task in the past years (Geiger et al., 2012; Grigorescu et al., 2020; Janai et al., 2020), we have arguably not yet reached the accuracy levels needed for safe operations in general (non-geo-fenced) settings.

One reason why 3D object detectors struggle is that often there simply is not enough information in the scene to make a good call. For example, the best LiDAR sensors may only yield a few tens of points on a small child if s/he is some distance away. From this very limited data, the 3D object detector must decide if this is a pedestrian who may rush onto the road at any moment, or if this is just a tree that can be ignored. Sophisticated machine learning models may bring all sorts of inductive biases to bear (Lang et al., 2019; Zhou & Tuzel, 2018; Yan et al., 2018; Shi et al., 2019; 2020a), but such few LiDAR points may fundamentally lack enough information to make an accurate decision (Kendall & Gal, 2017). What we need is additional information about the scene.

Current 3D object detectors treat every scene as completely novel and unknown — ignoring potentially valuable information from previous traversals of the same route. In fact, many of us drive through the same routes every day: to and from work, schools, shops, and friends. Even when we embark on a completely new route, we are often following in the footsteps of *other drivers* who have gone through this very section of a route before. In this paper we explore the following question:

> If we collect and aggregate *unlabeled* LiDAR information over time, potentially across vehicles, can we utilize the *past traversals of the same route* to better detect cars, bicycles and pedestrians?

We provide an *affirmative* answer to the above question. Our intuition is as follows: while historical traversals through a route will be different, i.e., we will encounter different cars on the road and

different pedestrians on the crosswalks, taken together, these traversals yield a wealth of information. For instance, they reveal where pedestrians, cars, and cyclists generally tend to be in the scene, and where a stop sign or some unknown background object is persistently present across traversals. With this contextual information at hand, a detector can better recognize, say, a pedestrian heavily occluded by cars on the roadside, since pedestrians have come and gone through these regions before, and it is unlikely that they are persistent background objects. Thus, leveraging such contextual information could substantially improve the detection accuracy in these safety-critical scenarios.

Concretely, we formalize this insight by proposing a simple and efficient approach that uses past traversals of a scene to vastly improve perception performance. Off-line, we use neural networks to digest past traversals of the scene into a sparse, compact, geo-referenced representation that we call SQuaSH (Spatial-Quantized Sparse History features). While in operation, a self-driving car can then query the local SQuaSH context of every LiDAR point in the current scene, thus enriching the information available to perform recognition. This information can be added as features to any LiDAR-based 3D object detector, and both the detector and the SQuaSH representation can be trained jointly *without any additional supervision*. The resulting detector is substantially more accurate, and in fact can become more accurate over time *without any retraining* as more traversals of the scene are collected. This local development and improvement with unlabeled data is an important distinction compared to other approaches, such as using high definition maps which are static and require extensive data collections and labeling.

We validate our approach on two large-scale, real-world self-driving datasets, Lyft Level 5 Perception (Kesten et al., 2019) and the nuScenes Dataset (Caesar et al., 2020), with multiple representative modern object detection models (Lang et al., 2019; Yan et al., 2018; Zhou & Tuzel, 2018; Shi et al., 2019; 2020a) under various settings and show consistent performance gains. Concretely, our contributions are three-fold:

1. We identify a rich trove of information in the form of *hindsight* from past traversals that can substantially boost 3D perception of challenging objects at no extra cost.

2. We propose a simple and efficient method to leverage such information without any extra labels, that can be incorporated into most modern 3D perception pipelines.

3. We evaluate our method on 3D object detection exhaustively across two large real-world datasets, different object types, and multiple detection architectures and demonstrate remarkably consistent and significant improvements, especially on the challenging cases — small, far-away objects — by over 300%.

## 2 RELATED WORKS

**3D object detection** is one of the most important perception tasks for autonomous driving. Most existing algorithms take LiDAR sensory data as input, which provides accurate 3D point clouds of the surrounding environment. There are two popular branches of methods. The first is to voxelize LiDAR point clouds and use the resulting 3D-cube representation as input to 2D or 3D convolutional neural networks to infer 3D bounding boxes (Zhou & Tuzel, 2018; Yan et al., 2018; Zhou et al., 2020; Li, 2017; Engelcke et al., 2017; Lang et al., 2019; Ku et al., 2018; Yang et al., 2018b; Liang et al., 2018; Chen et al., 2017; Shi et al., 2020a). The second is to design neural network architectures explicitly for point cloud inputs (Qi et al., 2018; 2017a;b; Shi et al., 2019; 2020b; Yang et al., 2020). Our approach augments the input point cloud with features queried from the history and is *agnostic* to specific object detection pipelines as long as they can take point clouds with per-point features. In this work, we experiment with four representative, high-performing 3D object detection models (subsection 4.1) and demonstrate significant and consistent improvements when using HINDSIGHT.

**3D object detection in contexts.** We propose to augment the raw point cloud captured at driving time by features queried from the previous traversals. This relates to a recent body of literature that augment point clouds with extra information, e.g., high-definition (HD) maps (Yang et al., 2018a; Ravi Kiran et al., 2018; Seif & Hu, 2016; Liang et al., 2020a) or semantic information from synchronized images (Chen et al., 2017; Qi et al., 2018; Ku et al., 2018; Liang et al., 2018; Wang et al., 2018; You et al., 2020; Vora et al., 2020). Our approach is orthogonal to – and compatible with – these methods, but also differs from them in key ways. In contrast to HD Maps, our history features are learned without additional labels, can be updated easily with newly collected traversals,

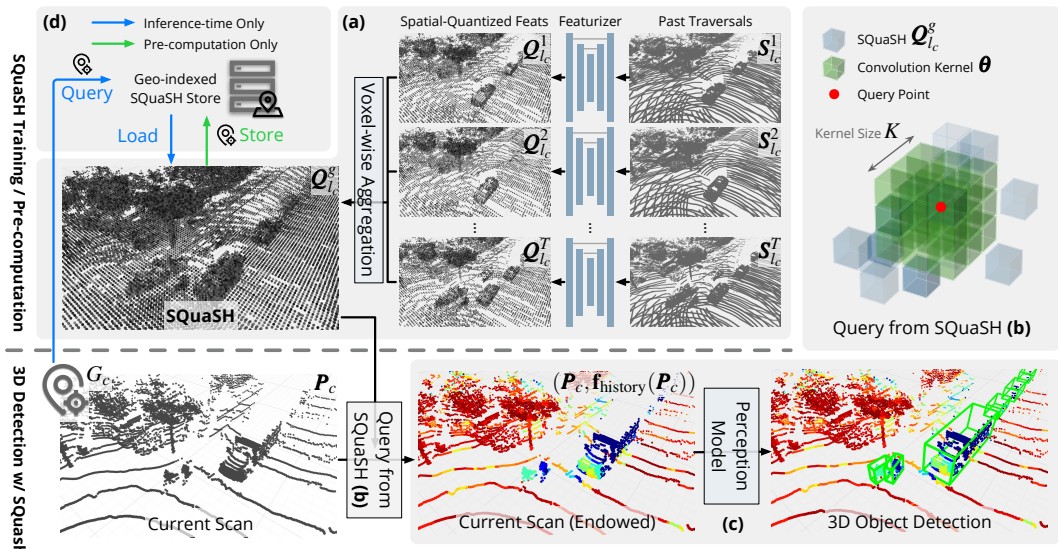

Figure 1: **Overall pipeline of HINDSIGHT.** The pipeline is divided into four parts: **(a)** SQuaSH computation; **(b)** endowing the LiDAR scan by querying from SQuaSH; **(c)** detection with the endowed scan; **(d)** pre-computing SQuaSH off-line for deployment. The coloring on points in the endowed scan (the bottom-center and -right figures) means we endow history features in $\mathbb{R}^{d_{\text{history}}}$ to each of the points. Please refer to section 3 for definitions of the symbols. Best viewed in color.

and are not limited to certain pre-defined information types. In contrast to images, querying our pre-computed history features adds infinitesimal latency to the perception pipeline (Table 3); the history features can disambiguate hard cases even if they are faraway from the ego car, whereas the corresponding image patches can be very small and hard to recognize.

**Perception with historical context.** There are works utilizing temporal scans as short-term context for perception (Liang et al., 2020b; Huang et al., 2020; Yang et al., 2021). However, there is very limited work on using *long-term* historical context for perception. A few works use videos of people interacting with a scene to discover the *affordances* in the scene, e.g., where people sit, stand, or walk (Delaitre et al., 2012; Fouhey et al., 2012; Sun et al., 2020). Closer in spirit to our work is the work by Barnes et al. (2018), which uses multiple traversals of a scene to map out regions that are *ephemeral*, namely, that represent transient objects. In these works, past observations are used to make specific, hand-crafted measurements. In contrast, our key idea is to automatically *learn* what is relevant from the history by training the SQuaSH representation end-to-end with the detector.

## 3 HINDSIGHT

**Problem setup.** We assume we are operating in a typical self-driving system: the autonomous vehicle is equipped with various synchronized sensors including LiDAR, which senses the surrounding scene by capturing a 3D point cloud, and GPS/INS, which provides accurate 6-Degree-of-Freedom (6-DoF, 3D displacement + 3D rotation) localization. The self-driving system reads the sensor inputs and detects other participants (cars, pedestrians and cyclists) in the scene. Most top-performing detection algorithms heavily rely on LiDAR scans for accurate detection. We denote the LiDAR point cloud captured at the current location during driving as $\boldsymbol{P}_c = \{(x_i, y_i, z_i)\}_{i=1}^k \in \mathbb{R}^{k \times 3}$, and denote the corresponding 6-DoF localization as $G_c$. We assume that transformed by $G_c$, $\boldsymbol{P}_c$ is in a fixed global 3D coordinate system. Atypical of this conventional setting, we assume we have access to *unlabeled* point cloud scans from past traversals through the same location.

**Overview of our approach HINDSIGHT.** In this work, we propose a novel approach, which we term HINDSIGHT, to endow the current scan $\boldsymbol{P}_c$ with information from past traversals to improve 3D detection (Figure 1). Given the current spatial location $G_c$, HINDSIGHT retrieves a set of point clouds recorded by past traversals of the same location, encodes them by a spatial featurizer, and aggregates them into a spatially-indexed feature grid (which we call SQuaSH, Figure 1 (a)). Then, for each point in the current scan $\boldsymbol{P}_c$, HINDSIGHT queries the local neighborhood of that point in

the feature grid, and linearly combines the features in this neighborhood to produce a "history" feature vector (Figure 1 (b)). This feature vector is concatenated as additional channels into the representation of each 3D point, producing an *endowed* point cloud. The endowed point cloud, enriched with the past information, is then fed into a 3D object detector (Figure 1 (c)). The whole pipeline is trained end-to-end with detection labels annotated only on current scans.

While this process seems time-consuming at first glance, many of the computations, especially those related to the past scans, can indeed be carried out off-line. Specifically, we deliberately separate the computation such that the feature grid SQuaSH from the past scans is pre-computed and stored in a geo-indexed manner (Figure 1 (d)). While in operation, the self-driving system only needs to query the pre-computed SQuaSH to endow the current point cloud with little latency.

Below, we describe each of these components in detail.

## 3.1 DENSE 3D DATA FROM PAST TRAVERSALS

We first describe the data we get from past traversals. We assume access to $T \geq 1$ past traversals; to ensure the information is up-to-date, we maintain the most recent $T_{\max}$ traversals. A single traversal $t$ consists of a sequence of point clouds $\{\boldsymbol{P}_f^t\}$ and the associated global localization $\{G_f^t\}$ recorded as the car was driven; here $f$ is the index of the frame. We transform each point cloud into a fixed global coordinate system. Then, at a location $l$ every $s$ metres along the road, we combine the point clouds from a range $[-H_s, H_e]$ to produce a *dense* point cloud $\boldsymbol{S}_l^t = \bigcup_{G_f^t \in [l-H_s, l+H_e]} \{\boldsymbol{P}_f^t\}$ (where with a slight abuse of notation, we have used $G_f^t$ for the 3D location where $f$ was captured).

Combining multiple scans in this way can densify the scene, providing us with detailed and high-resolution point clouds for regions faraway from the current position. A static bush in a distance can yield very sparse LiDAR points similar to that of a car, causing a false positive detection. However, it can be disambiguated with multiple past scans that have dense points, recorded when the car was much closer to the bush in past traversals. These dense scans thus provide context for our method to "see" further into the distance or around occlusions.

## 3.2 SPATIAL-QUANTIZED SPARSE HISTORY FEATURES (SQUASH)

Next we take each dense point cloud $\boldsymbol{S}_l^t$ from each traversal $t$ and encode it using a spatial featurizer that yields a spatially-quantized feature tensor $\boldsymbol{Q}_l^t$. In dense form, $\boldsymbol{Q}_l^t$ can be viewed as a 4D tensor in $\mathbb{R}^{H \times W \times D \times d_{\text{history}}}$, where $H$, $W$, and $D$ are the nominal 3D spatial dimensions and $d_{\text{history}}$ is the feature dimension. The quantization step size is $\delta$, so the point located at $(x, y, z)$ will be encoded at $(\lfloor x/\delta \rfloor, \lfloor y/\delta \rfloor, \lfloor z/\delta \rfloor)$. Any number of architectures can be used to map point clouds to such voxelized tensors. In this work, we adopt the Sparse Residual U-Net (SR-UNet) (Choy et al., 2019) as the spatial featurizer due to its high efficiency and performance. Additionally, because most parts of the scene are unoccupied, we represent $\boldsymbol{Q}_l^t$ as a sparse tensor.

For every location $l$ we compute $T$ such tensors, one for each traversal. We aggregate them into a single tensor $\boldsymbol{Q}_l^g$ by applying a per-voxel aggregation function $f_{\text{agg}}$:

$$\boldsymbol{Q}_l^g = f_{\text{agg}}(\boldsymbol{Q}_l^1, \ldots, \boldsymbol{Q}_l^T). \tag{1}$$

There are a number of available choices for the aggregation function, e.g., max-pooling, mean-pooling, *etc*. We do not observe a significant performance difference among them (possibly due to the small $T_{\max}$) and thus use max-pooling due to its simplicity. We term the aggregated tensor $\boldsymbol{Q}_l^g$ for location $l$ as the Spatial-Quantized Sparse History tensor (SQuaSH tensor) for this location.

## 3.3 QUERY FROM SQUASH

Suppose now that the self-driving car captures a new scan $\boldsymbol{P}_c$ at location $G_c$, and suppose that the SQuaSH tensor at this location is $\boldsymbol{Q}_{l_c}^g$. We endow each of the points in $\boldsymbol{P}_c$ by querying their surrounding features in $\boldsymbol{Q}_{l_c}^g$. To do so, we convolve $\boldsymbol{Q}_{l_c}^g$ with a $K \times K \times K$ filter $\boldsymbol{\theta}$, and look up each point $\boldsymbol{p} = (x, y, z)$ from the current scan in the resulting tensor after appropriate quantization:

$$\boldsymbol{F}_{l_c} = \boldsymbol{\theta} * \boldsymbol{Q}_{l_c}^g, \tag{2}$$

$$\mathbf{f}_{\text{history}}(\boldsymbol{p}) = \boldsymbol{F}_{l_c}[\lfloor x/\delta \rfloor, \lfloor y/\delta \rfloor, \lfloor z/\delta \rfloor], \tag{3}$$

where $\mathbf{f}_{\text{history}}(\boldsymbol{p})$ is the corresponding history feature for point $\boldsymbol{p}$. In practice, this entire operation can be easily implemented by a 3D sparse convolution layer in Minkowski Engine (Choy et al., 2019) and runs extremely fast on a GPU (Table 3).

## 3.4 3D DETECTION WITH SQuaSH

Most modern LiDAR-based detection algorithms are able to take this point cloud $\boldsymbol{P}_c$ with per-point features, e.g., LiDAR intensity. To make use of these off-the-shelf, high-performing detection algorithms without heavy modifications to their frameworks, we adopt the simplest way to incorporate the queried history features: attaching $\mathbf{f}_{\text{history}}(\boldsymbol{P}_c)$ as per-point features. Even with this simple strategy, we observe a significant and consistent performance gain. Our entire pipeline is simply trained in an end-to-end manner, including the spatial featurizer, the query filter $\boldsymbol{\theta}$ and the detection model, via minimizing the corresponding detection loss from the detection algorithms on a dataset.

## 3.5 DEPLOYMENT

For deployment purposes, the self-driving system generates the SQuaSH tensors $\boldsymbol{Q}_l$ for all locations $l$ in an off-line manner once the spatial featurizer is trained. The SQuaSH tensors can then be retrieved at run-time using a simple geo-location query. We intentionally disentangle the computation between the past traversals and the current scan so that such pre-computation is possible. This allows us to use large and complex spatial featurizers to process the past traversals *without introducing latency into the on-board autonomous driving system*. Moreover, these features can be stored efficiently, since the SQuaSH tensors are sparse and can be produced only once every $s = 10$ meters without losing performance (Appendix B). As mentioned before, obtaining history features from the retrieved SQuaSH is extremely fast, adding little burden to on-board systems (Table 3).

## 3.6 IMPLEMENTATION DETAILS

**Localization.** With current localization technology, we can reliably achieve very high localization accuracy (e.g., *1-2 cm-level* accuracy with RTK[1], *10 cm-level* with Monte Carlo Localization scheme (Chong et al., 2013) as adopted in (Caesar et al., 2020)). We assume an accurate localization within at least 30 cm-level, and show our method is robust to such localization error (Table 6).

**Spatial featurizer.** We use a SR-UNet inplemented in Minkowski Engine (Choy et al., 2019) as the spatial featurizer. It is a 14-layer U-Net (Ronneberger et al., 2015) following the conventional ResNet (He et al., 2016) basic block design with sparse 3D conv-/deconvolutions. The SR-UNet was originally designed by Choy et al. (2019) and achieved a significant improvement against previous methods on various challenging semantic segmentation benchmarks. In this work, we adopt this architecture to process each past traversal, and aggregate them by max-pooling at each quantized coordinate index into a SQuaSH tensor. In Table 4, we include an ablation study on the relation between the final detection performance and the complexity of the spatial featurizer.

**Hyper-parameters.** We set $[-H_s, H_e]$, the range of scans that are combined to produce dense point clouds, to be $[0, 20]$ m since we experiment with frontal-view detection only, and we combine only one scan into the dense point cloud $\boldsymbol{S}_l^t$ every 5 m within this range. When performing inference, we use the geographically closest possible SQuaSH tensor: we observe no performance drop if the available SQuaSH is within 5 m (Appendix B). We use default quantization size $\delta = 0.3$ m for all the experiments (except we use 0.4 m for PointPillars due to limited GPU memory in training), and we conduct an ablation study on the quantization sizes in Table 3. Due to GPU memory limit in training, we use $T_{\max} = 5$ available past traversals to compute the SQuaSH. The filter kernel size $K$ is 5. The dimensionality of the history features $d_{\text{history}}$ is 64.

## 4 EXPERIMENTS

### 4.1 SETUP

**Dataset.** We validate our approach, HINDSIGHT, on the Lyft Level 5 Perception Dataset (Kesten et al., 2019) (Lyft) and the nuScenes Dataset (Caesar et al., 2020). To the best of our knowledge,

---

[1]https://en.wikipedia.org/wiki/Real-time_kinematic_positioning

Table 1: **Detection performance of four detectors with and without HINDSIGHT on the Lyft Dataset.** We break down $AP_{BEV}$ by depth ranges. "+ H" stands for methods with point clouds endowed with history features by HINDSIGHT; "$\Delta$ $AP_{BEV}$" indicates the absolute gain. HINDSIGHT improves the baselines in all but one case; on several challenging cases (i.e., far-away objects or pedestrians and cyclists), the exact gains are larger than 20% in AP. Corresponding $AP_{3D}$ results and results under other IoU metrics are included in Appendix A where we observe a similar trend.

| Method | Car | | | | Pedestrian | | | | Cyclist | | | |
|---|---|---|---|---|---|---|---|---|---|---|---|---|
| | 0-30 | 30-50 | 50-80 | 0-80 | 0-30 | 30-50 | 50-80 | 0-80 | 0-30 | 30-50 | 50-80 | 0-80 |
| PointPillars | 77.5 | 67.6 | 42.8 | 64.4 | 40.2 | 22.1 | 2.1 | 17.2 | 53.7 | 21.3 | 1.6 | 29.5 |
| PointPillars + H | 78.6 | 72.0 | 51.7 | 68.9 | 65.5 | 49.1 | 17.0 | 43.6 | 62.3 | 46.3 | 5.0 | 46.6 |
| $\Delta$ $AP_{BEV}$ | +1.1 | +4.4 | +8.9 | +4.5 | +25.3 | +27.0 | +14.9 | +26.4 | +8.6 | +25.0 | +3.4 | +17.1 |
| SECOND | 76.2 | 66.2 | 40.6 | 62.9 | 45.4 | 27.2 | 6.7 | 22.1 | 74.0 | 33.6 | 1.8 | 38.9 |
| SECOND + H | 76.9 | 72.3 | 52.3 | 68.7 | 55.6 | 45.9 | 26.2 | 42.1 | 68.2 | 44.1 | 12.4 | 49.1 |
| $\Delta$ $AP_{BEV}$ | +0.7 | +6.1 | +11.7 | +5.8 | +10.2 | +18.7 | +19.5 | +20.0 | -5.8 | +10.5 | +10.6 | +10.2 |
| PointRCNN | 74.0 | 72.3 | 45.9 | 65.1 | 40.7 | 29.5 | 5.7 | 22.1 | 69.7 | 37.9 | 1.0 | 44.8 |
| PointRCNN + H | 75.2 | 77.5 | 56.2 | 70.7 | 61.8 | 53.1 | 19.3 | 43.7 | 69.7 | 47.3 | 7.2 | 53.0 |
| $\Delta$ $AP_{BEV}$ | +1.2 | +5.2 | +10.3 | +5.6 | +21.1 | +23.6 | +13.6 | +21.6 | +0.0 | +9.4 | +6.2 | +8.2 |
| PV-RCNN | 77.3 | 71.9 | 43.2 | 65.8 | 38.5 | 16.6 | 4.8 | 14.9 | 63.2 | 38.2 | 3.9 | 37.0 |
| PV-RCNN + H | 80.8 | 77.4 | 56.4 | 73.5 | 44.6 | 42.0 | 24.9 | 35.5 | 63.2 | 47.7 | 12.7 | 48.1 |
| $\Delta$ $AP_{BEV}$ | +3.5 | +5.5 | +13.2 | +7.7 | +6.1 | +25.4 | +20.1 | +20.6 | +0.0 | +9.5 | +8.8 | +11.1 |

these two datasets are the only two publicly available, large-scale autonomous driving datasets that have both annotated object bounding-box labels and multiple traversals with accurate localization. Since they are not originally designed for evaluating the efficacy of utilizing historical traversals, some samples (i.e., point clouds at specific timestamps) in the datasets do not have past traversals. Thus, we re-split the datasets so that each training and test sample has at least 2 past traversals; sequences in training and test set are *geographically disjoint*. This results in 12,407/2,274 training/test samples in the Lyft dataset and 3,985/2,324 training/test samples in the nuScenes dataset.

To apply off-the-shelf 3D object detectors which are typically built to take KITTI data (Geiger et al., 2012), we convert the raw Lyft and nuScenes data into the KITTI format and only train and evaluate frontal-view, frame-by-frame, and LiDAR-only detections. We use the roof LiDAR signal (40 or 64 beams in Lyft; 32 beams in nuScenes), and obtain the global 6-DoF localization and extrinsic transformations between LiDAR and GPS/IMU directly from the raw data.

**Evaluation metric.** We follow KITTI (Geiger et al., 2012) and evaluate object detection both in the bird's-eye view (BEV) and in 3D for *Car, Pedestrian,* and *Cyclist* on the Lyft dataset (Cyclist includes Bicycle + Motorcycle in the raw Lyft dataset), and *Car* and *Pedestrian* in the nuScenes dataset, since there are too few cyclist instances. We also follow the KITTI convention to report average precision (AP) with the intersection over union (IoU) thresholds at 0.7/0.5 for Car and 0.5/0.25 for Pedestrian and Cyclist. We denote AP for BEV and 3D by $AP_{BEV}$ and $AP_{3D}$. Since the Lyft and nuScenes datasets do not provide the official definition of object difficulty as that in KITTI, we follow Wang et al. (2020) to evaluate the AP at various depth ranges. In Tables 1 to 5 of the main paper, due to the limited space, we only present results with IoU=0.7 for car objects and IoU=0.5 for pedestrian and cyclist objects. Please refer to Appendix A for results on $AP_{3D}$ and other IoU metrics, where we observe a similar trend.

**Base 3D object detectors.** We adopt four representative, high-performing 3D object detectors: PointPillars (Lang et al., 2019), SECOND (VoxelNet) (Yan et al., 2018; Zhou & Tuzel, 2018), PointRCNN (Shi et al., 2019), and PV-RCNN (Shi et al., 2020a), as the base detector models to augment (and compare) with our HINDSIGHT framework. These models are all state-of-the-art LiDAR-based detectors and are distinct in architecture: pillar-wise (PointPillars) *vs*. voxel-wise (SECOND, PV-RCNN) *vs*. point-wise (PointRCNN) features, and single-stage (PointPillars, SECOND) *vs*. multi-stage (PointRCNN, PV-RCNN). We use publicly available code from Open-PCDet (OpenPCDet, 2020) for all four models. We use most of the default hyper-parameters tuned for KITTI, with the exception that we enlarge the perception range from 70m to 90m – since Lyft provides labels beyond 70m – and we reduce the number of training epochs by 1/4 on Lyft – since its train set is about three times of the size of KITTI. We observe consistent results with such hyper-parameters. We use exactly the same hyper-parameters and detection model architectures for the

Table 2: **Detection performance of two detectors with and without HINDSIGHT on the nuScenes Dataset.** Please refer to Table 1 for naming. HINDSIGHT improves the baselines in all but one case; the gain is more pronounced on the challenging pedestrian objects.

| Method | Car | | | | Pedestrian | | | |
|---|---|---|---|---|---|---|---|---|
| | 0-30 | 30-50 | 50-80 | 0-80 | 0-30 | 30-50 | 50-80 | 0-80 |
| PointPillars | 24.5 | 4.2 | 0.0 | 11.7 | 13.8 | 1.4 | 0.0 | 6.3 |
| PointPillars + H | 26.6 | 6.1 | 0.1 | 13.5 | 21.8 | 3.5 | 0.0 | 11.1 |
| $\Delta$ AP$_{\text{BEV}}$ | +2.1 | +1.9 | +0.1 | +1.8 | +8.0 | +2.1 | +0.0 | +4.8 |
| PointRCNN | 25.8 | 5.1 | 0.8 | 13.0 | 23.5 | 1.2 | 0.0 | 10.7 |
| PointRCNN + H | 28.5 | 7.3 | 1.5 | 14.7 | 33.9 | 4.5 | 1.4 | 17.3 |
| $\Delta$ AP$_{\text{BEV}}$ | +2.7 | +2.2 | +0.7 | +1.7 | +10.4 | +3.3 | +1.4 | +6.6 |

Table 3: **Detection performance with various quantization sizes with PointRCNN.** We show AP$_{\text{BEV}}$ / AP$_{\text{3D}}$ evaluated in 0-80m. "H-$\delta$ m" stands for HINDSIGHT with quantization size $\delta$ m. We also report the average SQuaSH storage size and the average latency of querying the SQuaSH per scene under different quantization sizes.

| Method | Query Latency | SQuaSH Size | Car | Ped. | Cyc. |
|---|---|---|---|---|---|
| PointRCNN | 0 ms | 0 MB | 65.1 / 41.8 | 22.1 / 16.8 | 44.8 / 36.0 |
| + H-0.2 m | 4.1 ms | 55.0 MB | 59.7 / 39.8 | 36.8 / 25.9 | 46.7 / 39.3 |
| + H-0.3 m | 3.4 ms | 32.3 MB | **70.7 / 46.3** | **43.7 / 33.4** | **53.0 / 46.8** |
| + H-0.5 m | 2.4 ms | 15.3 MB | 68.0 / 44.6 | 40.9 / 29.1 | 45.2 / 38.2 |
| + H-1.0 m | 1.8 ms | 5.2 MB | 61.3 / 36.7 | 32.4 / 23.2 | 42.1 / 34.1 |

baseline and baseline + HINDSIGHT, except a change in the input dimension of the first layer due to the endowed hindsight features. We experiment with all four models on Lyft, and PointPillars and PointRCNN on nuScenes, as training diverged for the other two models with the above default hyperparameters. All models are trained with 4 NVIDIA 3090 GPUs.

## 4.2 EMPIRICAL RESULTS

### 4.2.1 3D OBJECT DETECTION RESULTS

In Table 1 and Table 2 we show the detection performance of the four representative 3D object detection models with and without the aid of HINDSIGHT on two different datasets and various object types. (Please refer to Appendix A for corresponding AP$_{\text{3D}}$ results and results under other IoU metrics.) We observe that HINDSIGHT consistently improves the detection performance for all detectors in almost all cases. In particular, models with HINDSIGHT enjoy substantial improvement over baseline counterparts on mid- to far-range, small (pedestrian and cyclist) objects, **frequently yielding accuracy gains of over 20 percentage points in AP**. As also shown in Figure 2 (more in Appendix C), compared to base detectors, HINDSIGHT maintains a significantly higher precision under the same recall for detection objects. We note that these are typically considered the *hardest* parts in LiDAR-based object detection, not only because small objects have much fewer training examples in these datasets, but also because small physical shapes and farther distance from the ego car inherently result in much sparser LiDAR signals. It is thus crucial to incorporate the historical prior of the scenes (captured by HINDSIGHT) for improved detection performance.

### 4.2.2 LATENCY AND STORAGE

Low latency is crucial for on-board perception models since they need to make critical decisions real-time. In Table 3, we examine the extra latency that HINDSIGHT introduces to the detection model. We observe an exceptionally low latency from our query operation (average 3.4 ms for quantization size 0.3 m). It adds little burden to the real-time pipeline whose perception loop typically takes 100 ms (10 Hz).

In Table 3 we also show the average uncompressed storage of one SQuaSH for a scene. Each SQuaSH roughly covers 90 m * 80 m range in BEV. The storage highly depends on the quantization size; it takes 32.3 MB for the default quantization size 0.3 m. Though it is already small enough in light of current cheaper and cheaper storage technology, in practice, the autonomous driving system

Table 4: **Detection performance of different HINDSIGHT spatial featurizer with PointR-CNN.** We show $AP_{BEV}$ / $AP_{3D}$ evaluated in 0-80m. "+H $(x)$" stands for HINDSIGHT with spatial featurizer $x$.

| Method | Car | Ped. | Cyc. |
|---|---|---|---|
| PointRCNN | 65.1 / 41.8 | 22.1 / 16.8 | 44.8 / 36.0 |
| + H (Identity) | 67.0 / 44.9 | 36.4 / 25.2 | 51.6 / 43.7 |
| + H (FCN) | 69.7 / 44.1 | 38.6 / 28.4 | 48.7 / 44.3 |
| + H (SR-UNet) | **70.7 / 46.3** | **43.7 / 33.4** | **53.0 / 46.8** |

Table 5: **Detection performance with various number of available past traversals with PointRCNN.** We show $AP_{BEV}$ / $AP_{3D}$ evaluated in 0-80m. "+H $(\leq x)$" stands for HINDSIGHT with using $\leq x$ past traversals.

| Method | Car | Ped. | Cyc. |
|---|---|---|---|
| PointRCNN | 65.1 / 41.8 | 22.1 / 16.8 | 44.8 / 36.0 |
| + H $(= 1)$ | 66.8 / 44.9 | 31.4 / 22.8 | 46.3 / 38.5 |
| + H $(\leq 2)$ | 70.0 / **46.3** | 42.7 / 29.0 | 52.9 / 43.5 |
| + H $(\leq 5)$ | **70.7 / 46.3** | **43.7 / 33.4** | **53.0 / 46.8** |

Table 6: **Detection performance with simulated localization error with PointRCNN.** We show $AP_{BEV}$ / $AP_{3D}$ evaluated in 0-80m. "H $\Delta\sigma$ m" means adding a random localization error of a standard deviation $\sigma$ m in evaluation.

| Method | Car | Ped. | Cyc. |
|---|---|---|---|
| PointRCNN | 65.1 / 41.8 | 22.1 / 16.8 | 44.8 / 36.0 |
| + H $\Delta$0.0 m | **70.7** / 46.3 | 43.7 / **33.4** | **53.0 / 46.8** |
| + H $\Delta$0.1 m | 70.6 / **46.8** | **44.0** / 31.8 | 52.9 / 45.4 |
| + H $\Delta$0.2 m | 70.3 / 43.6 | 42.7 / 30.4 | 52.5 / 43.3 |
| + H $\Delta$0.3 m | 70.1 / 44.2 | 37.9 / 26.0 | 49.4 / 41.0 |
| + H $\Delta$0.5 m | 67.2 / 40.8 | 31.0 / 20.8 | 44.5 / 36.3 |
| + H $\Delta$1.0 m | 60.9 / 33.8 | 16.7 / 12.3 | 30.1 / 24.2 |

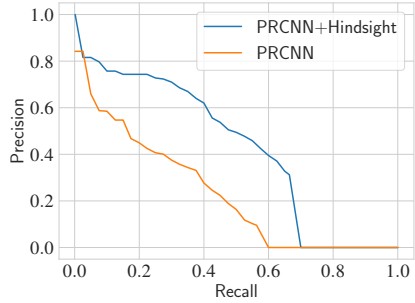

Figure 2: **Precision-recall curves for *Pedestrian* of PointRCNN with and without HINDSIGHT**, under metric BEV IoU=0.5 in 0-80m.

can build the SQuaSH at least every 10 metres without losing performance (Appendix B) to have a storage-efficient engineering implementation.

### 4.2.3 ABLATION STUDY

**Impact of the HINDSIGHT spatial featurizer.** In Table 4 we study the relation between the complexity of HINDSIGHT spatial featurizer and the final detection performance. We compare three possible spatial featurizers with different complexities: Identity, which directly quantizes each historical traversal and uses nominal "1" as features in $Q_l^t$ to indicate the non-emptiness; FCN, which is a simple two-layer, fully convolutional network; SR-UNet, the model we use as default. It is interesting to observe that by simply querying from the history occupancy tensor (Identity), HINDSIGHT enjoys a reasonable performance boost; the detection model benefits even more with more complex feature backbones (FCN and SR-UNet). This suggests that even the simple presence or absence of objects in past historical traversals is a valuable signal. More sophisticated processing of the past traversals may also characterize exactly what objects were observed in the past, which is richer contextual information.

**Impact of the amount of available past traversals.** In Table 5 we study the effect of using different numbers of past traversals during testing. We observe that with just one past traversal, the detection model can already benefit from the history features; with only two, the model can achieve performance close to using five traversals (our manually-selected maximum). This result indicates that HINDSIGHT does not need a large number of past traversals to extract relevant information.

**Impact of quantization sizes.** The spatial quantization sizes of SQuaSH can affect the granularity and the receptive field of HINDSIGHT. With a fixed spatial featurizers architecture, the smaller the quantization size is, the higher the granularity but the smaller the receptive field is. In Table 3, we study how different quantization sizes affect the final detection performance. It can be seen that the performance gain is more pronounced with a proper quantization size (0.3m). When the quantization is too small, the model cannot capture sufficient surrounding priors and is susceptible to random errors in localization; when the quantization size is too large, HINDSIGHT loses its ability to differentiate points within one voxel and it becomes an extra burden to the learning process.

**Impact of localization error.** HINDSIGHT relies on (accurate) localization to query the correct history features. Thus, it is important to investigate the robustness of HINDSIGHT against potential

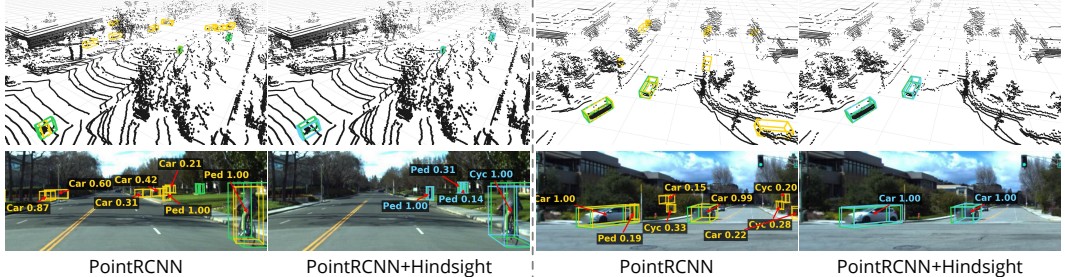

|              |                      |              |                      |
|:------------:|:--------------------:|:------------:|:--------------------:|
| PointRCNN    | PointRCNN+Hindsight  | PointRCNN    | PointRCNN+Hindsight  |

Figure 3: **Qualitative visualization of detection results.** We visualize two randomly picked scenes in the Lyft test set, and plot the detection results from the base detector (PointRCNN) with and without HINDSIGHT on LiDARs (upper) and images (bottom, not used in detection). Ground-truth bounding boxes are shown in green, base and HINDSIGHT detections are shown in yellow and cyan. Detection classes and confidence scores are attached. Zoom in for details. Best viewed in color.

localization error. In Table 6, we take the model trained with ground-truth localization (directly obtained from the dataset, without correcting for already present, minor errors), and evaluate its performance with the presence of simulated random localization error. We simulate the error by sampling a unit vector in 3D space , scale it by $\epsilon \sim \mathcal{N}(0, \sigma^2)$, where $\sigma$ is the error magnitude, and apply it to the test scan. We observe that HINDSIGHT is robust to localization error, where it can still bring substantial improvement on detection performance for all noise less than $\sigma = 0.3\,(\mathrm{m})$. Such localization error is well within the reliable range of current localization technology, showing the superior real-world application of HINDSIGHT. We also test the model under the presence of random bearing error (please refer to Appendix D), where we observe similar robustness.

### 4.3 QUALITATIVE RESULTS

**Detection performance.** In Figure 3, we visualize the detection results of PointRCNN and PointR-CNN+HINDSIGHT at two random scenes of the Lyft test set (please refer to Appendix F for more). We observe that the base detector tends to generate many false positives to ensure a good coverage of the true objects; our HINDSIGHT has significantly fewer false positives while being able to detect the true objects exhaustively (also illustrated by Figure 2). Specifically, for faraway objects (two distant pedestrians in the left scene) and small objects (the close cyclist in the left scene), the base detector either completely misses them or misclassifies them, while HINDSIGHT works like a charm.

## 5 DISCUSSION

**Limitations.** Our approach relies on the past traversals being informative of the current state of the scene. Therefore, sudden changes to the current scene (e.g., a snow-storm or an accident) may pose problems to the detector, since the scene it observes will be inconsistent with the history. Note that such adverse conditions may impact traditional detectors as well. Unfortunately, existing datasets are collected in relatively mild conditions (e.g., clear weather for Lyft and sunny, rainy, or cloudy conditions for nuScenes). As such, they do not offer the opportunity to test our approach under adverse weather or other abnormal changes to road conditions. We hope this work can facilitate future research on this topic and inspire the creation of more related diverse datasets.

**Conclusion.** In this work, we explore a novel yet highly practical scenario for self-driving, where unlabeled LiDAR scans from previous traversals of the same scene are available. We propose HIND-SIGHT, which extracts the contextual information from multiple past traversals and aggregates them into a compact and geo-indexed SQuaSH. While the self-driving system is in operation, SQuaSH can be used to enrich the current 3D scan and substantially improve 3D perception. The whole pipeline can be trained end-to-end with detection labels only. Once trained, the SQuaSH can be pre-computed off-line and queried on-line with little latency. We show on two large-scale, real-world self-driving datasets that HINDSIGHT consistently and substantially improves the detection performance of four high-performing 3D detectors, taking us one step closer towards safe self-driving. With this work, we hope to inspire more research into exploring the rich trove of information from past traversals of the same scene.

## ACKNOWLEDGEMENT

This research is supported by grants from the National Science Foundation NSF (III-1618134, III-1526012, IIS-1149882, IIS-1724282, TRIPODS-1740822, IIS-2107077, OAC-2118240, OAC-2112606 and IIS-2107161), the Office of Naval Research DOD (N00014-17-1-2175), the DARPA Learning with Less Labels program (HR001118S0044), the Bill and Melinda Gates Foundation, the Cornell Center for Materials Research with funding from the NSF MRSEC program (DMR-1719875), and SAP America.

## ETHICS STATEMENT

This work uses data from past traversals of a scene to substantially improve 3D perception, especially for far-off or occluded objects. We believe that our approach will lead to safer self-driving cars. Our approach will also allow for self-driving cars to be driven beyond tight geo-fenced areas where HD maps are available, thus making this technology more broadly accessible.

Because our approach stores data that is typically discarded, a potential concern is the privacy of pedestrians, cars and cyclists on the road. However, we note that the data we store is heavily quantized ($\delta = 0.3m$), featurized and only based on the sparse LiDAR point cloud. Therefore the stored representation is likely to contain only aggregate information about the presence or absence of certain kinds of objects, and is unlikely to have any identifying information.

## REPRODUCIBILITY STATEMENT

In subsection 3.6 and subsection 4.1, we exhaustively list the hyper-parameters for our approach, the dataset details, evaluation metrics and the hyper-parameters for training 3D object detectors. Our code is available at https://github.com/YurongYou/Hindsight. As mentioned in subsection 4.1, since the Lyft and the nuScenes dataset are not originally designed for evaluating the efficacy of utilizing historical traversals, we use a custom split to have a valid evaluation. We also release the split in the link for future comparison.

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

# Appendix

We provide details omitted in the main text.

- Appendix A: detailed evaluation results on the Lyft and the nuScenes dataset. (subsubsection 4.2.1 of the main paper, same naming as in Table 1 and Table 2)
- Appendix B: ablation on using pre-computed SQuaSH from a distance.
- Appendix C: more precision-recall curves of detection results (as in Figure 2).
- Appendix D: evaluation under random bearing error.
- Appendix E: evaluation results of Table 3 under different metrics.
- Appendix F: more qualitative visualization (as in subsection 4.3).

## A  MORE EVALUATION RESULTS ON THE LYFT DATASET AND THE NUSCENES DATASET

In Table 1 and Table 2 of the main paper, we present evaluation results across various depth ranges on $AP_{BEV}$ with IoU=0.7 for car objects and IoU=0.5 for pedestrian and cyclist objects, on both the Lyft and the nuScenes dataset. In Tables 7 to 12, we present the evaluation results on $AP_{BEV}$ and $AP_{3D}$ with two sets of IoU metrics (IoU=0.7 for car objects and IoU=0.5 for pedestrian and cyclist objects; IoU=0.5 for car objects and IoU=0.25 for pedestrian and cyclist objects).

| Method | Car | | | | Pedestrian | | | | Cyclist | | | |
|---|---|---|---|---|---|---|---|---|---|---|---|---|
| | 0-30 | 30-50 | 50-80 | 0-80 | 0-30 | 30-50 | 50-80 | 0-80 | 0-30 | 30-50 | 50-80 | 0-80 |
| PointPillars | 85.9 | 77.4 | 51.0 | 74.0 | 47.1 | 27.0 | 3.4 | 21.0 | 64.1 | 29.5 | 3.9 | 38.0 |
| PointPillars + H | 87.9 | 83.1 | 64.4 | 80.2 | 73.9 | 60.0 | 21.5 | 52.4 | 66.5 | 49.8 | 11.9 | 52.9 |
| $\Delta$ $AP_{BEV}$ | +2.0 | +5.7 | +13.4 | +6.2 | +26.8 | +33.0 | +18.1 | +31.4 | +2.4 | +20.3 | +8.0 | +14.9 |
| SECOND | 87.3 | 75.8 | 47.9 | 72.4 | 58.2 | 36.5 | 9.9 | 29.8 | 81.4 | 44.7 | 7.1 | 47.7 |
| SECOND + H | 88.0 | 83.2 | 66.2 | 80.8 | 72.6 | 61.1 | 37.3 | 56.3 | 77.6 | 53.7 | 24.4 | 59.6 |
| $\Delta$ $AP_{BEV}$ | +0.7 | +7.4 | +18.3 | +8.4 | +14.4 | +24.6 | +27.4 | +26.5 | -3.8 | +9.0 | +17.3 | +11.9 |
| PointRCNN | 86.8 | 78.0 | 58.1 | 76.9 | 50.9 | 36.4 | 7.5 | 27.7 | 74.5 | 48.4 | 2.8 | 51.8 |
| PointRCNN + H | 86.9 | 83.1 | 72.2 | 82.8 | 79.5 | 63.3 | 26.2 | 54.7 | 74.1 | 54.2 | 13.5 | 59.5 |
| $\Delta$ $AP_{BEV}$ | +0.1 | +5.1 | +14.1 | +5.9 | +28.6 | +26.9 | +18.7 | +27.0 | -0.4 | +5.8 | +10.7 | +7.7 |
| PV-RCNN | 86.3 | 78.0 | 49.7 | 74.3 | 47.6 | 24.3 | 6.8 | 20.4 | 65.9 | 42.8 | 6.6 | 41.0 |
| PV-RCNN + H | 86.5 | 85.1 | 67.0 | 81.7 | 56.4 | 52.3 | 32.9 | 45.9 | 69.9 | 55.3 | 24.1 | 55.7 |
| $\Delta$ $AP_{BEV}$ | +0.2 | +7.1 | +17.3 | +7.4 | +8.8 | +28.0 | +26.1 | +25.5 | +4.0 | +12.5 | +17.5 | +14.7 |

Table 7: HINDSIGHT detection results in the Lyft dataset, in $AP_{BEV}$ with IoU=0.5 for car objects and IoU=0.25 for pedestrian and cyclist objects.

| Method | Car | | | | Pedestrian | | | | Cyclist | | | |
|---|---|---|---|---|---|---|---|---|---|---|---|---|
| | 0-30 | 30-50 | 50-80 | 0-80 | 0-30 | 30-50 | 50-80 | 0-80 | 0-30 | 30-50 | 50-80 | 0-80 |
| PointPillars | 58.3 | 34.9 | 13.1 | 36.7 | 28.9 | 14.0 | 1.4 | 11.7 | 39.5 | 15.1 | 0.2 | 20.3 |
| PointPillars + H | 59.1 | 33.1 | 16.0 | 37.4 | 43.2 | 29.9 | 13.8 | 28.2 | 54.1 | 34.7 | 1.9 | 38.8 |
| $\Delta$ $AP_{BEV}$ | +0.8 | -1.8 | +2.9 | +0.7 | +14.3 | +15.9 | +12.4 | +16.5 | +14.6 | +19.6 | +1.7 | +18.5 |
| SECOND | 60.3 | 36.5 | 15.5 | 39.1 | 30.9 | 16.7 | 4.3 | 13.9 | 59.2 | 27.0 | 0.6 | 30.7 |
| SECOND + H | 60.3 | 39.3 | 16.4 | 41.0 | 36.1 | 31.0 | 19.5 | 27.5 | 61.8 | 29.7 | 3.3 | 37.1 |
| $\Delta$ $AP_{BEV}$ | +0.0 | +2.8 | +0.9 | +1.9 | +5.2 | +14.3 | +15.2 | +13.6 | +2.6 | +2.7 | +2.7 | +6.4 |
| PointRCNN | 60.0 | 42.2 | 17.9 | 41.8 | 33.0 | 22.3 | 3.9 | 16.8 | 59.7 | 26.5 | 0.3 | 36.0 |
| PointRCNN + H | 59.5 | 48.0 | 24.0 | 46.3 | 53.3 | 40.4 | 13.0 | 33.4 | 66.1 | 39.5 | 2.9 | 46.8 |
| $\Delta$ $AP_{BEV}$ | -0.5 | +5.8 | +6.1 | +4.5 | +20.3 | +18.1 | +9.1 | +16.6 | +6.4 | +13.0 | +2.6 | +10.8 |
| PV-RCNN | 64.3 | 46.2 | 18.0 | 45.2 | 25.0 | 11.3 | 4.2 | 10.4 | 58.4 | 30.4 | 1.4 | 30.9 |
| PV-RCNN + H | 67.1 | 46.1 | 21.4 | 46.8 | 33.2 | 29.8 | 17.7 | 25.7 | 53.8 | 32.3 | 5.5 | 34.9 |
| $\Delta$ $AP_{BEV}$ | +2.8 | -0.1 | +3.4 | +1.6 | +8.2 | +18.5 | +13.5 | +15.3 | -4.6 | +1.9 | +4.1 | +4.0 |

Table 8: HINDSIGHT detection results in the Lyft dataset, in $AP_{3D}$ with IoU=0.7 for car objects and IoU=0.5 for pedestrian and cyclist objects.

| Method | Car | | | | Pedestrian | | | | Cyclist | | | |
|---|---|---|---|---|---|---|---|---|---|---|---|---|
| | 0-30 | 30-50 | 50-80 | 0-80 | 0-30 | 30-50 | 50-80 | 0-80 | 0-30 | 30-50 | 50-80 | 0-80 |
| PointPillars | 85.0 | 73.5 | 46.0 | 70.6 | 47.1 | 26.9 | 3.4 | 21.0 | 64.1 | 29.2 | 3.8 | 37.8 |
| PointPillars + H | 85.7 | 77.3 | 58.6 | 75.6 | 73.0 | 60.0 | 21.5 | 51.8 | 66.5 | 49.8 | 11.2 | 52.0 |
| $\Delta$ $AP_{BEV}$ | +0.7 | +3.8 | +12.6 | +5.0 | +25.9 | +33.1 | +18.1 | +30.8 | +2.4 | +20.6 | +7.4 | +14.2 |
| SECOND | 85.6 | 72.7 | 43.8 | 70.1 | 57.3 | 36.3 | 9.9 | 29.7 | 81.4 | 43.6 | 6.9 | 47.2 |
| SECOND + H | 85.9 | 80.2 | 59.0 | 76.5 | 72.3 | 61.1 | 37.3 | 56.2 | 76.3 | 53.1 | 22.8 | 58.6 |
| $\Delta$ $AP_{BEV}$ | +0.3 | +7.5 | +15.2 | +6.4 | +15.0 | +24.8 | +27.4 | +26.5 | -5.1 | +9.5 | +15.9 | +11.4 |
| PointRCNN | 86.7 | 75.9 | 53.9 | 74.7 | 50.2 | 35.5 | 7.5 | 27.4 | 74.5 | 48.4 | 1.8 | 51.4 |
| PointRCNN + H | 86.8 | 82.7 | 64.9 | 80.1 | 78.0 | 62.9 | 26.2 | 54.6 | 74.1 | 54.3 | 13.3 | 59.5 |
| $\Delta$ $AP_{BEV}$ | +0.1 | +6.8 | +11.0 | +5.4 | +27.8 | +27.4 | +18.7 | +27.2 | -0.4 | +5.9 | +11.5 | +8.1 |
| PV-RCNN | 86.2 | 76.1 | 45.5 | 71.3 | 46.1 | 24.3 | 6.8 | 20.3 | 66.3 | 42.8 | 6.3 | 40.8 |
| PV-RCNN + H | 86.4 | 81.1 | 60.2 | 78.8 | 54.9 | 52.3 | 32.9 | 45.8 | 69.9 | 54.3 | 22.5 | 55.9 |
| $\Delta$ $AP_{BEV}$ | +0.2 | +5.0 | +14.7 | +7.5 | +8.8 | +28.0 | +26.1 | +25.5 | +3.6 | +11.5 | +16.2 | +15.1 |

Table 9: HINDSIGHT detection results in the Lyft dataset, in $AP_{3D}$ with IoU=0.5 for car objects and IoU=0.25 for pedestrian and cyclist objects.

| Method | Car | | | | Pedestrian | | | |
|---|---|---|---|---|---|---|---|---|
| | 0-30 | 30-50 | 50-80 | 0-80 | 0-30 | 30-50 | 50-80 | 0-80 |
| PointPillars | 31.9 | 6.9 | 0.1 | 16.0 | 19.4 | 3.1 | 0.0 | 9.7 |
| PointPillars + H | 34.4 | 10.2 | 0.1 | 18.3 | 34.9 | 8.0 | 0.1 | 18.8 |
| $\Delta$ $AP_{BEV}$ | +2.5 | +3.3 | +0.0 | +2.3 | +15.5 | +4.9 | +0.1 | +9.1 |
| PointRCNN | 29.0 | 7.1 | 1.5 | 15.6 | 31.8 | 3.4 | 0.3 | 15.4 |
| PointRCNN + H | 33.9 | 10.9 | 1.8 | 18.8 | 47.5 | 10.2 | 3.0 | 25.7 |
| $\Delta$ $AP_{BEV}$ | +4.9 | +3.8 | +0.3 | +3.2 | +15.7 | +6.8 | +2.7 | +10.3 |

Table 10: HINDSIGHT detection results in the nuScenes dataset, in $AP_{BEV}$ with IoU=0.5 for car objects and IoU=0.25 for pedestrian objects.

| Method | Car | | | | Pedestrian | | | |
|---|---|---|---|---|---|---|---|---|
| | 0-30 | 30-50 | 50-80 | 0-80 | 0-30 | 30-50 | 50-80 | 0-80 |
| PointPillars | 11.5 | 0.7 | 0.0 | 4.8 | 10.8 | 0.6 | 0.0 | 4.7 |
| PointPillars + H | 11.5 | 1.3 | 0.0 | 5.0 | 14.9 | 1.5 | 0.0 | 6.9 |
| $\Delta$ $AP_{BEV}$ | +0.0 | +0.6 | +0.0 | +0.2 | +4.1 | +0.9 | +0.0 | +2.2 |
| PointRCNN | 15.5 | 0.9 | 0.1 | 6.7 | 17.6 | 0.7 | 0.0 | 7.8 |
| PointRCNN + H | 16.3 | 2.0 | 0.1 | 7.5 | 26.1 | 2.9 | 0.4 | 13.2 |
| $\Delta$ $AP_{BEV}$ | +0.8 | +1.1 | +0.0 | +0.8 | +8.5 | +2.2 | +0.4 | +5.4 |

Table 11: HINDSIGHT detection results in the nuScenes dataset, in $AP_{3D}$ with IoU=0.7 for car objects and IoU=0.5 for pedestrian objects.

| Method | Car | | | | Pedestrian | | | |
|---|---|---|---|---|---|---|---|---|
| | 0-30 | 30-50 | 50-80 | 0-80 | 0-30 | 30-50 | 50-80 | 0-80 |
| PointPillars | 28.7 | 4.9 | 0.0 | 13.6 | 18.8 | 2.8 | 0.0 | 9.3 |
| PointPillars + H | 31.7 | 7.3 | 0.1 | 16.0 | 34.3 | 6.8 | 0.0 | 18.0 |
| $\Delta$ $AP_{BEV}$ | +3.0 | +2.4 | +0.1 | +2.4 | +15.5 | +4.0 | +0.0 | +8.7 |
| PointRCNN | 28.2 | 5.5 | 1.1 | 14.0 | 31.3 | 3.1 | 0.3 | 15.3 |
| PointRCNN + H | 31.7 | 8.9 | 1.3 | 16.8 | 47.1 | 10.2 | 3.0 | 25.5 |
| $\Delta$ $AP_{BEV}$ | +3.5 | +3.4 | +0.2 | +2.8 | +15.8 | +7.1 | +2.7 | +10.2 |

Table 12: HINDSIGHT detection results in the nuScenes dataset, in $AP_{3D}$ with IoU=0.5 for car objects and IoU=0.25 for pedestrian objects.

Table 13: **Detection performance using SQuaSH pre-computed at various distance from the current ego car pose.** We use base detector PointRCNN and report $AP_{BEV}$ with IoU=0.7 for car objects and IoU=0.5 for pedestrian and cyclist objects. We do not observe significant performance difference among using SQuaSH with $0-5$ m offset.

| Method | Car | Ped. | Cyc. |
|---|---|---|---|
| PointRCNN | 65.1 / 41.8 | 22.1 / 16.8 | 44.8 / 36.0 |
| + H-SQuaSH  0    m offset | 70.7 / 46.3 | 43.7 / 33.4 | 53.0 / 46.8 |
| + H-SQuaSH  2.5 m offset (backward) | 70.1 / 44.2 | 45.0 / 31.6 | 55.2 / 46.1 |
| + H-SQuaSH  2.5 m offset (forward) | 63.0 / 44.8 | 43.3 / 31.8 | 53.6 / 46.0 |
| + H-SQuaSH  5.0 m offset (backward) | 70.0 / 46.7 | 44.5 / 32.8 | 53.1 / 45.5 |
| + H-SQuaSH  5.0 m offset (forward) | 70.3 / 46.3 | 43.6 / 32.6 | 53.8 / 45.0 |
| + H-SQuaSH 10.0 m offset (backward) | 70.1 / 46.2 | 45.0 / 33.5 | 55.4 / 48.0 |
| + H-SQuaSH 10.0 m offset (forward) | 70.2 / 46.3 | 36.5 / 29.1 | 52.9 / 45.3 |

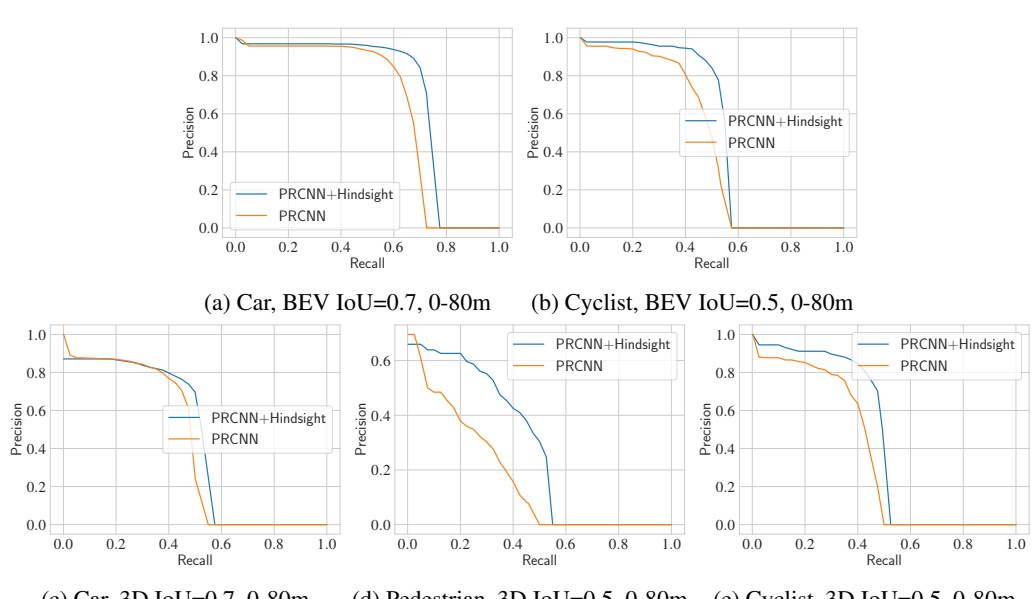

(a) Car, BEV IoU=0.7, 0-80m     (b) Cyclist, BEV IoU=0.5, 0-80m

(c) Car, 3D IoU=0.7, 0-80m     (d) Pedestrian, 3D IoU=0.5, 0-80m     (e) Cyclist, 3D IoU=0.5, 0-80m

Figure 4: **P-R curves for various objects of PointRCNN model with and without HINDSIGHT.**

## B  SQUASH AT A DISTANCE

For experiments in the main paper, we use the SQuaSH pre-computed at the closest possible location from the ego car position $G_c$ (i.e., the $l_c$ is picked as the closest possible one from the ego car). For practical use, the self-driving system should pre-compute the SQuaSH once every $s$ m to have a storage-efficient implementation. In this case, the available SQuaSH can have $s/2$ m offset from the ego car localization. We evaluate this situation in Table 13, where we use the SQuaSHs that are $s/2$ m away (either backward or forward) from the ego car localization. We observe no significant performance difference in $s/2 = 0-5$ m offset. It validates that in practice the self-driving system can pre-compute the SQuaSH at least every 10 m without losing performance gains.

## C  ADDITIONAL PRECISION-RECALL CURVES

In Figure 4, we show the detection precision-recall curves of the base detector (PointRCNN) and the base detector with HINDSIGHT, on car, pedestrian and cyclist objects under various evaluation metrics. We observe a similar trend as shown in Figure 2.

Table 14: **Detection performance with simulated bearing error with PointRCNN.** We show $AP_{BEV}$ / $AP_{3D}$ evaluated in 0-80m. "H $\Delta\sigma°$" means adding a random bearing error of a standard deviation $\sigma°$ in evaluation.

| Method | Car | Ped. | Cyc. |
|---|---|---|---|
| PointRCNN | 65.1 / 41.8 | 22.1 / 16.8 | 44.8 / 36.0 |
| + H $\Delta 0.0°$ | 70.7 / 46.3 | 43.7 / 33.4 | 53.0 / **46.8** |
| + H $\Delta 0.1°$ | **70.8 / 46.8** | **47.8 / 34.9** | **53.7** / 45.5 |
| + H $\Delta 0.5°$ | 70.1 / 46.6 | 35.8 / 24.8 | 48.2 / 41.0 |
| + H $\Delta 1.0°$ | 69.9 / 44.3 | 25.0 / 16.8 | 44.7 / 37.8 |

Table 15: **Detection performance with various quantization sizes with PointRCNN.** We show $AP_{BEV}$ / $AP_{3D}$ evaluated in 0-80m. Please refer to Table 3 for naming. We show the evaluation under Car: IoU 0.5, Ped & Cyc: IoU 0.25.

| Method | Car | Ped. | Cyc. |
|---|---|---|---|
| PointRCNN | 76.9 / 74.7 | 27.7 / 27.4 | 51.8 / 51.4 |
| + H-0.2 m | 78.0 / 72.8 | 50.0 / 50.0 | 55.9 / 55.7 |
| + H-0.3 m | **82.8 / 80.1** | **54.7 / 54.6** | **59.5 / 59.5** |
| + H-0.5 m | 80.0 / 77.6 | 51.2 / 51.2 | 51.3 / 51.2 |
| + H-1.0 m | 72.2 / 69.7 | 41.5 / 41.1 | 45.8 / 45.8 |

## D  ADDITIONAL BEARING ERROR EVALUATION

Typically, with current technology, the error on bearing angle is much smaller than that on localization (as small as 0.08 degrees[2]). Nevertheless, we evaluate the performance under a much harsher estimate of bearing rotation errors, and present detection evaluation results in Table 14. It can be seen that even under situations with large (potentially overestimated) bearing angle errors, HINDSIGHT models still significantly outperform the baseline model.

## E  ADDITIONAL EVALUATION RESULTS WITH DIFFERENT QUANTIZATION

We show the evaluation results of Table 3 under a different metric (car object: IoU 0.5, pedestrian & cyclist object: IoU 0.25) in Table 15.

## F  ADDITIONAL QUALITATIVE RESULTS

We show additional qualitative detection results in Figure 5.

---

[2]https://hexagondownloads.blob.core.windows.net/public/Novatel/assets/Documents/Papers/PwrPak7D-E1-PS/PwrPak7D-E1-PS.pdf

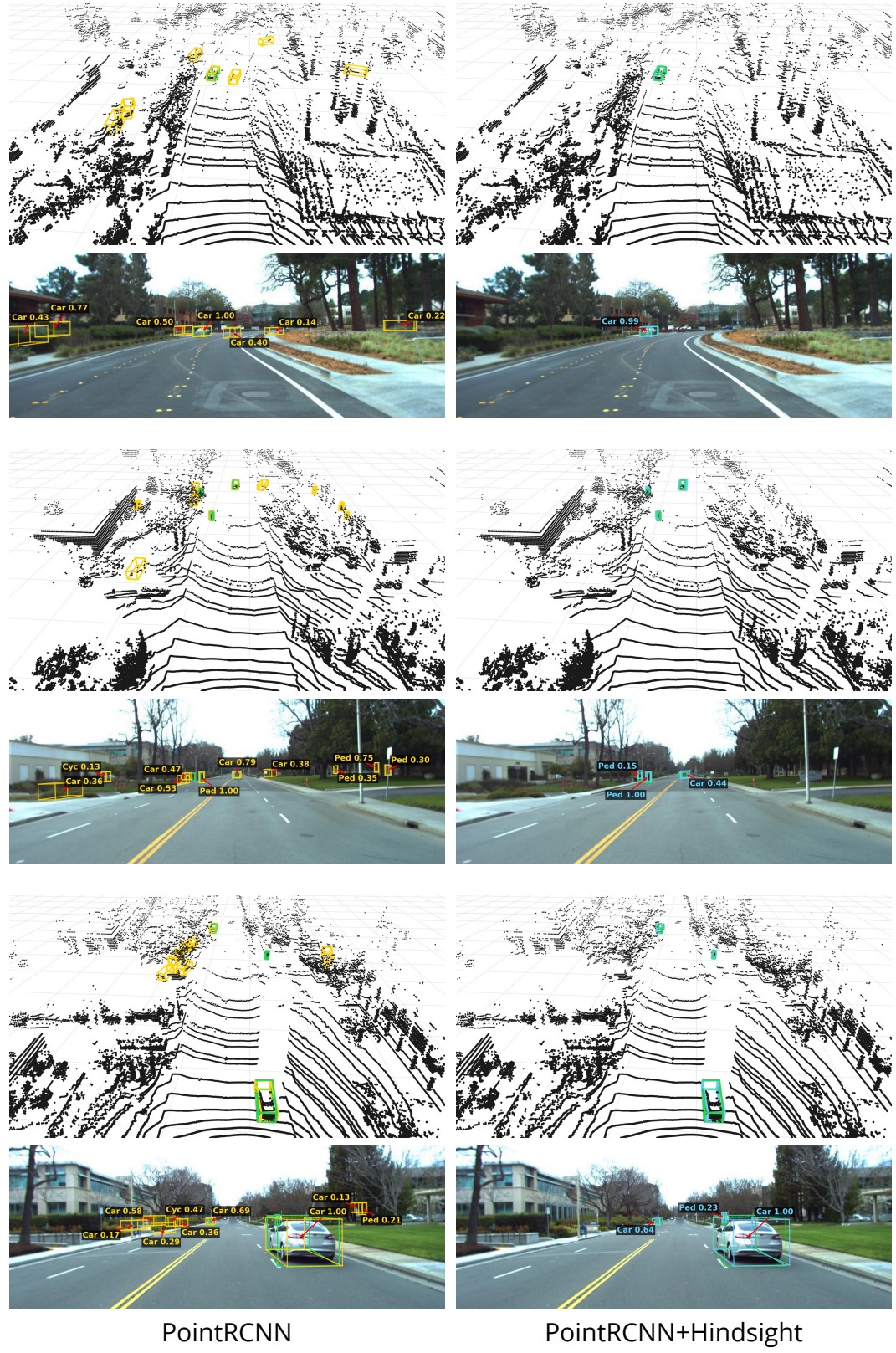

Figure 5: **Qualitative visualization of detection results.** Please refer to Figure 3 for coloring.

