# OpenReview forum: "Hindsight is 20/20: Leveraging Past Traversals to Aid 3D Perception"
_ICLR.cc/2022/Conference — ICLR 2022 Poster_

### Official Review · Reviewer_i3iC · 2021-11-01

**Correctness:** 3
**Technical Novelty And Significance:** 3
**Empirical Novelty And Significance:** 3
**Recommendation:** 6
**Confidence:** 4

**Main Review:**


# Pros:

+ The overall idea is a good one. Leveraging these past point clouds, which would typically be thrown away, is sensible, and the algorithm is a good and simple approach to using these past traversals, which effectively give a kind of 'peak ahead and behind' into areas of the scene not currently seen well. As SLAM and relocalisation algorithms improve, the algorithm will be more and more suited to different scenarios.

+ I like that fact that the authors used four different object detection algorithms to show the gains are independent of detection method used.
Experiments are generally well thought out and well presented. A good set of experiments in different scenarios are shown, using two sensible datasets.

* Including qualitative figures helps a little to understand the algorithms improvements.

* The paper is very well presented, with clear figures and tables and nice typesettings

* English is good and explanations are clear.

* While related work feels a little short it gives a very clear overview of related works, categorising them nicely.


# Cons:

- My main concern with the paper is a lack of introspection. In the introduction, the authors say that past traversals "...reveal where pedestrians, cars, and cyclists generally tend to be in the scene, and where a stop sign or some unknown background object is persistently present across traversals. With this contextual information at hand, a detector can better recognize, say, a pedestrian heavily occluded by cars on the roadside, since pedestrians have come and gone through these regions before, and it is unlikely that they are persistent background objects." But this is speculation; I was really hoping for (and expecting!) an experiment which confirmed this, shedding some light on what the network is actually learning, and how it is really helping improve scores. For example the 'pedestrian' class undergoes a surprisingly large increase (Table 5, Fig 2).

- Some ideas for things that might help give introspection include:

1) Looking at the split of improvements on moving vs static objects.

2) Does detection rate only improve in regions where previous traversals actually contained instances of the object being detected? Or does the detection rate also improve where previous traversals contained no cars/pedestrians/cyclists?

3) Some of the following experiments might also help:

- I'd be keen to see an extra row in e.g. Table 5, which uses one 'past traversal', but where that past traversal is in fact just the entire current traversal. So the SQuaSH features would have access to information about the scene captured e.g. 5 or 10 seconds before and after the current scan P_c.

- Similarly, I'd like to see an extra row in Table 5 where the 'past traversal' is a copy of the current scan *but only up to the time at which P_c was captured*.

- It might not be easy, but: I'd have really liked to see a baseline which uses some kind of SOTA recurrent network to make use of the point clouds which immediately came before P_c temporally. And/or merging a short time history of pointclouds together into one single point cloud for input to the detection algorithms. (It feels a little unfair that the baseline only has access to point data at a single timepoint, when models already exist to make use of time series data). See e.g. the LSTM works I list below.

- Related work feels a little short.

  - There are more 'change detection' works which could be worth mentioning. [City-scale Scene Change Detection using Point Clouds, ICRA 2021] is a recent example I found.

  - In terms of "Perception with historical context" – it might be worth mentioning that *short-term* historical context is very often used where temporal data is present, e.g. in an LSTM-based model. It is just longer-term historical context which is more rarely used. For example:

    - Liang et al. PnPNet: End-to-End Perception and Prediction with Tracking in the Loop, CVPR 2020
    - Huang et al. An LSTM Approach to Temporal 3D Object Detection in LiDAR Point Clouds. ECCV 2020
    - Yang et al. 3D-MAN: 3D Multi-frame Attention Network for Object Detection. CVPR 2021.



**Summary Of The Paper:**

 The paper proposes a method for improving object detection in lidar scans, which works in cases where a lidar scanner visits the same environment multiple times. The paper proposes to take point clouds from previous traversals and put them through a trainable network, before temporally aggregating features from each of the traverals. (It is assumed that camera pose is known for each scan). The aggregated features are then concatenated with the features from the current traversal, before being input into an off-the-shelf object detector. Results show that this method improves object detection rates in almost all cases, regardless of the object classification method used (the authors run experiments with four different object detectors).



**Summary Of The Review:**

Overall I would like to see this paper in ICLR.

It is a nice idea, a well-written paper, and I would learn something from it. There are lots of results in the paper, but most of them are simply showing that the algorithm improves scores. I would really like to see a couple more experiments helping to show *why* the scores improve. I would also like to see some stronger baselines using point cloud data immediately preceding P_c as additional input.

---

> ### Author Response · Authors · 2021-11-18
> **Thank you for your detailed and constructive review!**
>
> Thank you for your detailed and constructive review! And we appreciate your support for our paper! Here are our brief replies to your questions:
> 1. Regarding introspection: Thanks for your suggestions! One of the ways we tried to quantitatively evaluate the gain was by looking at the precision-recall curve of different classes (Figure 2 and Figure 4). We found that Hindsight generates significantly much less false positive detection, especially for the pedestrian class. We do not observe the model has a significant difference on detecting moving/static objects (here we mean car, pedestrian and cyclist). We will look more into the detection results in the coming week and will update if we have new findings.
> 2. For additional experiments:
>     1. In regards to “past traversal is the current traversal”: we were hesitant to have this be a baseline, since it is not accurately portraying a realistic self-driving setting. During a normal drive, we do not have access to data from the future; thus we consider this an “oracle” that is not physically feasible. If the goal behind this experiment suggestion is to validate if only one traversal can help, we actually have included an ablation study (Table 5) where we only use 1 past traversal, also showing a decent improvement against the baseline.
>     2. In regards to “copy of current scan up to the current time”: this is an interesting direction for doing detection with temporal information, but we think it is probably out of the scope of this paper. We focus on validating the value of **long-term history** for 3D perception, which is orthogonal to detection with temporal information. And note that, this will require running SQuaSH computation online (since the ''SQuaSH'' for the latest temporal information cannot be cached beforehand), which will significantly impact the latency. We do believe that the way we integrate the history and current information (i.e., computing SQuaSH and query from it) can also be applied to other tasks, like detection with temporal information as you mentioned, and we are keen to see more follow-up works on this.
>    3. In regards to the "SOTA recurrent network": thanks for the suggestions, but unfortunately there is no official open-source code for the works the reviewer pointed to and we were not able to easily reproduce the experiments within a short period of time. In the paper, we have exhaustively tested out four high-performing detectors on two different large-scale datasets to validate the generalizability of our approach. We are happy to potentially include more in the future if the open-source code is available.
> 3. About the related work: Thanks for pointing out these papers! In terms of “historical context”, we meant utilizing data from the same environment in a past, different traversal (long-term history) and thus we are not sure if models using temporal information will fit in this topic. Specifically, the works cited use past context within the same traversal, whereas we utilize a tangential source of information (we use information from different traversals of the same location). Hindsight should be compatible with each of the LSTM-based models as long as they take point-wise features and we do not claim that Hindsight should replace them. We will clarify this in the final version.

---

> > ### Comment · Reviewer_i3iC · 2021-11-29
> > **Follow-up**
> >
> > Thank a lot for your reply and for addressing all of my thoughts!
> >
> > 1. Yes I think the PR curves are a little helpful. I think with some more thought on this I am still happy for the paper to be published without  additional introspection. After all, methods like monocular depth estimation were around for some time before any 'introspection' work was done (e.g. How Do Neural Networks See Depth in Single Images?, ICCV 2019)
> >
> > 2. Ok – I understand the reasons why you can't do some of these experiments. I think the oracle ones though would still be very helpful. I agree that experiments (1) and (2) as you list them aren't 'true' testing scenarios, but I think they would help to tell us the extent to which it is (a) the information in the previous scans which helps, vs (b) the additional network/alternative embedding which helps.
> >
> > 3. I agree your approach is different to LSTM methods, but I still think short-term memory papers are *related* and worthy of at least a sentence or two.

---

> > > ### Author Response · Authors · 2021-11-29
> > > **Thanks a lot for your reply!**
> > >
> > > Thanks a lot for your reply!
> > > - Regarding your insight on introspection: we completely agree with you that there should be more in-depth and meta-experiments to analyze the proposed model, and we will look into this and update the final version.
> > > - For the "oracles" experiments: we understand that they are intended to ablate whether (a) the information in the previous scans or (b) the network structure helps more. We think it is a great question! But we are concerned that these experiments may introduce additional information (from the future and/or the recent past) into the model which might not give us a clear conclusion to this question. Instead, we tried to answer this question from the perspective of varying the complexity of the network structure (Table 4), where with the exact same data, we used 1) Identity (0% complexity) 2) FCN 3) SR-UNet (100% complexity) as the spatial featurizer and analyzed the performance difference. We will definitely keep looking into this.
> > > - For LSTM methods: we agree with you that, while these works take a different approach than our method, they are definitely related to this topic! We will add them to the related work in the final version.
> > >
> > > In light of our clarifications, we would like to ask if you are willing to reconsider your score, and also if there are any additional questions we can respond to!

---

### Official Review · Reviewer_TGjG · 2021-11-01

**Correctness:** 4
**Technical Novelty And Significance:** 4
**Empirical Novelty And Significance:** 4
**Recommendation:** 8
**Confidence:** 4

**Main Review:**

## Strengths

  - [S1] The proposed technique seems to substantially and
    consistently improve perception performance without the
    need for any additional labels, or much additional
    online compute. This has been shown on two datasets
    with different LiDARs - nuScenes (32-beam) and Lyft L5
    (64-beam).
  - [S2] In-depth analysis and detailed discussion of
    Hindsight's performance. The module is shown to help on
    a wide range of base models - PointPillars, PointRCNN,
    SECOND.
  - [S3] The authors mention they plan to open source the
    code upon paper acceptance, which would be very
    beneficial to the research community.
  - [S4] The writing is good and the main ideas of the
    paper are presented clearly. For the few areas where
    clarity could be improved I made corresponding notes in
    the "Suggestions" section.
  - [S5] Detailed ablation studies, including an assessment
    of the impact of localization error on the approach
    (i.e., what happens if the current LiDAR is misaligned
    with the sweeps from the past?).


## Limitations

  - [L1] While HD maps can be more expensive than the
    proposed approach, it is still necessary in my opinion
    to compare against models which use them (I believe at
    least nuScenes does come with HD maps). This can help
    put the benefits of the current paper into perspective.
    Moreover, HD map creation itself can be automated to a
    substantial extent - things like BEV semantic maps,
    lane boundaries, drivable regions, etc. can all be
    computed without human intervention. Furthermore,
    whether or not using HD maps is better than the
    proposed approach would not weaken the paper. If HD
    maps are better, then the proposed method is still
    simpler and less expensive. If HD maps are not as good,
    then that's one more pro for the current method.
  - [L2] The proposed method requires prior passes through
    the same area, even at validation time. Even though
    these prior validation passes do not require labels, it
    still means that the benefits of the model only come
    after one, two, or more journeys have already been
    completed through an area. (Note that this limitation
    holds for HD maps as well.)
  - [L3] (Minor) The proposed approach shares some of the
    limitations of HD maps. For example, past observations
    may become outdated as the world changes - previous
    dangerous hidden driveways could disappear, but they
    would still appear in the "memories" causing confusion,
    potentially. However, the authors do touch on this in
    the "Limitations" section.


## Suggestions (Per-Section)

### Introduction

  - The setting is motivated overall very well, but I would
    adjust the statement on "Current 3D object detectors"
    to hint about HD maps. HD maps are discussed later in
    the section, but I believe already mentioning them in
    this paragraph can make the introduction a bit more
    clear.


### Method

  - In Figure 1, adding colors to the spatial-quantized
    feats and SQuaSH images could make it even clearer that
    you are talking about feature point clouds, not just
    the geometry. (It could be a different color scheme
    from (c) to differentiate the offline features from the
    endowed online point cloud.)
  - Q1: What is the 'e' from $H_e$?


### Experimental Results

  - Thank you for providing brief details on the new split
    as well as a recap of the metrics you are using. This
    makes the paper more self-contained and easier to
    follow.
  - Q2: What is the size of the physical area stored? What
    area does, e.g., the 55MB of SQuaSH in Table 3 cover?
    How does this scale for the entire operational domain
    of an SDV? It would be more informative in my opinion
    to show something like SQuaSH size / km of road, so
    that one may estimate the storage requirements for a
    given city/neighborhood.
  - Q3: Does the performance gap between H-0.2m and 0.3m
    depend on the IoU threshold for the metric? I find it
    odd that the difference for the car class is so large.
    Showing this analysis for different IoU thresholds
    could potentially help shed more light on the topic.
  - Q4: What do you mean by the need to "build SQuaSH"
    every K meters? Aren't the SQuaSH features essentially
    a continuous (sparse) tensor covering the entire
    operational domain? Can't this tensor simply be queried
    using the online detector RoI at arbitrary positions?
  - Q5: Do any of the evaluated detectors take multiple
    LiDAR sweeps as input? If yes, in Section 4.2 is the
    same localization offset applied to all sweeps?
    (Localization errors tend to be highly correlated
    temporally.)
  - (Very Minor) The citation key for OpenPCDet is a bit
    odd - is it possible to override it so that Bibtex does
    not try to split "OpenPCDet Team" into a first and a
    last name?

## Video/Supplementary

  The supplementary contains additional qualitative and
  quantitative results such as $\text{AP}_\text{3D}$ metrics and
  additional ablation studies.


**Summary Of The Paper:**

  The paper tackles the task of object detection in LiDAR
  point clouds, with a strong emphasis on automotive
  applications.

  The proposed approach is orthogonal to model, labeling,
  or training improvements, and consists in retrieving
  unlabeled geo-referenced past data from the area in which
  the car is located at inference time in order to augment
  the present observations. This is shown to have a very
  significant impact in perception quality, especially for
  smaller objects, such as pedestrians, and at longer
  range.

  Chunks of unlabeled data from the same geographic area as
  the input frames are pre-processed by a sparse 3D fully
  convolutional neural net, and the resulting features are
  used as additional inputs to the primary object detector.

  This pre-processor, which is called the "hindsight
  encoder" net, is trained jointly with the detector, as it
  is straightforward to backpropagate through the
  concatenation of "hindsight features" with the original
  point cloud. Since the hindsight features are essentially
  just retrieved by indexing and concatenated with the
  original input point cloud before it is fed into an
  arbitrary object detector, they can be used and trained
  in conjunction with a wide range of detector
  architectures. Once training finishes, they can be
  pre-computed offline to avoid inference-time overhead.

  Given the method's reliance on past (unlabeled)
  experiences through the area being evaluated, the authors
  re-partition the datasets on which they evaluate in order
  to match this setting (e.g., validation contains
  unlabeled data which is just used to compute the
  hindsight features for the labelled validation logs).
  While this makes it a bit difficult to compare against
  existing results, the authors run multiple baselines in
  their setting and promise to release the split
  information to the public. The proposed setting (lots of
  unlabeled data from existing passes through the
  "inference" area) is very common in the self-driving
  industry.

  The paper presents very promising results on Lyft and
  nuScenes, especially for pedestrians, cyclists, and
  distant objects (>50m) in general. The writing is clear
  and most key design choices are evaluated using ablation
  studies.


**Summary Of The Review:**

  Apart from a few minor concerns, such as comparisons to
  the use of "traditional" HD maps, I do not see any major
  flaws with the proposed paper. The proposed technique is
  orthogonal to improvements in architecture, data
  augmentation, etc., and it seems to work in a wide range
  of settings. The paper itself is well written and the
  experimental evaluation is thorough.

---

> ### Author Response · Authors · 2021-11-18
> **Thank you for your very detailed and constructive review!**
>
> Thank you for your very detailed and constructive review! And we appreciate your support for our paper! In particular, we appreciate your per-section suggestions and will incorporate them into the final version of our manuscript. Here are our brief replies to your questions and concerns:
> 1. [L1]: To our knowledge, HDNet [1] is the only published 3D object detection model that is designed to take in HDMap as input. But unfortunately, it is not open-sourced and we were not able to easily reproduce the experiments in the paper. However, we need to point out that we did not intend to claim that Hindsight should replace HDMap, and indeed we think Hindsight could serve as additional information that is quite different from HDMap to achieve safe driving.
> 2. For intro and the citation: thanks for your suggestions! We will modify them accordingly in the final version. For Figure 1, we tried coloring SQuaSH in our earlier version, but we could not get a visually clean figure and ended up with our current version.
> 3. The ‘e’ in H_e is from “end” (and the ‘s’ in H_s is from “start”).
> 4. Q2: each SQuaSH covers about 90m * 80m range in BEV. Thanks for the suggestion! We will change accordingly.
> 5. Q3: We show the evaluation on Car: IoU 0.5, Ped, Cyc: IoU 0.25 here. We will include it in the supplementary in the final version.
> 	| AP_BEV / AP_3D |     Car     |     Ped.    |     Cyc.    |
> |:-----------------------------------------------:|:-----------:|:-----------:|:-----------:|
> |                    PointRCNN                    | 76.9 / 74.7 | 27.7 / 27.4 | 51.8 / 51.4 |
> |                    + H-0.2 m                    | 78.0 / 72.8 | 50.0 / 50.0 | 55.9 / 55.7 |
> |                    + H-0.3 m                    | 82.8 / 80.1 | 54.7 / 54.6 | 59.5 / 59.5 |
> |                    + H-0.5 m                    | 80.0 / 77.6 | 51.2 / 51.2 | 51.3 / 51.2 |
> |                    + H-1.0 m                    | 72.2 / 69.7 | 41.5 / 41.1 | 45.8 / 45.8 |
> 6. Q4: You are correct that SQuaSH can be a continuous sparse tensor with proper engineering implementation. We choose to build SQuaSH every K meter in evaluation because it is easier to implement while proving the effectiveness of Hindsight.
> 7. Q5: No, for consistency we evaluate only detectors with single sweep LiDAR.
>
> [1] Bin Yang, Ming Liang, and Raquel Urtasun. Hdnet: Exploiting hd maps for 3D object detection. In CoRL, 2018a.

---

> > ### Comment · Reviewer_TGjG · 2021-11-25
> > **Follow-up**
> >
> > Thank you for the detailed response! The paper looks good - looking forward to the final version!

---

### Official Review · Reviewer_iXT8 · 2021-11-03

**Correctness:** 4
**Technical Novelty And Significance:** 4
**Empirical Novelty And Significance:** 4
**Recommendation:** 8
**Confidence:** 3

**Main Review:**

This is a well-written paper and a good contribution to the field. AVs often repeat the same route over and over again and this work is a step in the direction of utilizing past information to improve object detection metrics along the route without any additional labeling and training requirement. This will save money, time and effort that would otherwise go into obtaining more data and manual labeling of objects along the route. There are good ablation studies done regarding the effect of localization error in terms of displacement, but what is the effect of bearing error in localization - and it would be good to have that as well. The choice of the Mikowski-net or SR-Unet with sparse 3D convolutions is obviously a significant contributor to the results. An obvious question that arises is how much would such a network contribute to the metrics if used as a backbone for the 3D object detection as well.
There is a typo on page 5: “deconvoluitons”.


**Summary Of The Paper:**

This paper aggregates LIDAR scans from past traversals of an AV route to improve 3D object detection. Multiple scans from past traversals are aggregated into a voxellized representation using an accurate GPS-RTK localization system. Each scan is then passed through a state-of-the-art sparse 3D convolution network, SR-UNet, which generates a per-voxel feature that is stored as a representation for that part of the route. Such representations, called SQUASH, are stored periodically along the route, and added to the current LIDAR scan during inference as an additional input to a standard LIDAR based object detection network. The addition of these historical features improves object recognition results by ~20 average-precision points on NuScenes and Lyft Level 5 datasets, without any additional training.


**Summary Of The Review:**

This is a well-written paper with a significant contribution towards using historical data from the route to aid in object detection along repeated routes.

---

> ### Author Response · Authors · 2021-11-18
> **Thank you for your constructive feedback and we appreciate your support!**
>
> Thank you for your constructive feedback and we appreciate your support!
> 1. We will correct the typos in the final version.
> 2. For the ablation test on bearing error: thanks for bringing this up. Typically, with current technology, the error on bearing angle is much smaller than that on localization (as small as 0.08 degrees, https://hexagondownloads.blob.core.windows.net/public/Novatel/assets/Documents/Papers/PwrPak7D-E1-PS/PwrPak7D-E1-PS.pdf) Nevertheless, we evaluate the performance under a much harsher estimate of bearing rotation errors, and here are the results (under Car: IoU 0.7, Ped, Cyc: IoU 0.5):
> | AP_BEV / AP_3D  | Car         | Ped.        | Cyc.        |
> |------------------------------------------------|-------------|-------------|-------------|
> | PointRCNN                                      | 65.1 / 41.8 | 22.1 / 16.8 | 44.8 / 36.0 |
> | + H ∆r 0 deg                                   | 70.7 / 46.3 | 43.7 / 33.4 | 53.0 / 46.8 |
> | + H ∆r 0.1 deg                                 | 70.8 / 46.8 | 47.8 / 34.9 | 53.7 / 45.5 |
> | + H ∆r 0.5 deg                                 | 70.1 / 46.6 | 35.8 / 24.8 | 48.2 / 41.0 |
> | + H ∆r 1.0 deg                                 | 69.9 / 44.3 | 25.0 / 16.8 | 44.7 / 37.8 |
>
>     It can be seen that even under situations with large (potentially overestimated) bearing angle errors, Hindsight models will still significantly outperform the baseline model.
> 3. For being a backbone for 3D object detection: we expect that Mikowski-net or SR-Unet with sparse 3D convolutions would be applicable as the backbone for a 3D object detector. Indeed, they have been used in indoor 3D detection and have shown promising results. However, we leave this open to future research as we view our work as a preliminary exploration into this novel problem setting.

---

> > ### Comment · Reviewer_iXT8 · 2021-11-25
> > **Follow-up**
> >
> > Thanks for the ablation study with orientation error in localization. Looking forward to the final paper.

---

### Official Review · Reviewer_9mnc · 2021-11-04

**Correctness:** 4
**Technical Novelty And Significance:** 4
**Empirical Novelty And Significance:** 4
**Recommendation:** 8
**Confidence:** 4

**Main Review:**

Strength:
[1] The paper echoes with a few previous literature in the insight that previous observation may help scene understanding, which is intuitive and yet under explored. In this sense, the formulation of the motivation as well as the pipeline is novel, and successful experiment has proven its value to the community. In particular, the pipeline is especially useful for pedestrian detection, which is a difficult among all detection tasks, and the performance have somewhat saturated in recent years. With the additive nature of the proposed method, it may provide boost when combined with any state-of-the-art detection pipelines.

[2] The model is simple yet effective. The pipeline takes an existing feature extraction model to acquire the offline feature of past traversals, but was able to prove the effectiveness of this simple strategy to summarize information from past observations. The addition of the history features to current scan is also simple, by simply fusion it with current feature, and keep the same detection framework.

[3] The experiments are well carried out; in particular extensive ablative study is included on both the model design and results, which shows the pipeline is easy to integrate with existing detectors, and brings little overhead with noticeable improvement.

Weakness:
[1] As mentioned earlier, the pipeline is simple in that it reuses existing feature extractor and detector. However the paper did not includes more insights into the design choices in this part: (a) what if the feature extractor and the detector can be finetuned together? (2) how will be the offline history feature be updated when more data are observed? What will be the procedure and the cost?

[2] Clarity. For example, in Sec. 3.1, how are the dense point cloud acquired from combining samples sparse ones? Is there an alignment step involved, or simply shift all points to center at the current location? Extra visualization/example on this would be helpful.

[3] Results. To demonstrate how pedestrian detection benefits most, the paper should include more results and analysis on the pedestrian case. For example, how does improvement in pedestrian detection quantitatively compare to improvement in cars? Will there be any difference between moving people and standing ones, in that fusion will be worse for moving people.

**Summary Of The Paper:**

The paper proposes a HINDSIGHT framework for object detection on lidar scans, with query of scene feature extracted offline from previous traversals. The pipeline is additive and complementary to any real-time detection pipeline, by introducing additional point-wise feature from offline feature structure to point feature of the current scan. The paper is able to report improved performance on standard detection tasks, and in particular boost on pedestrian detection where past observation can help differentiate transient objects. Complete ablation study is included; the experiments are well-designed and described.

**Summary Of The Review:**

The major strength of the paper is the formulation of the problem and the insights it brings to this classic task of object detection in autonomous driving domain. The models are well designed, experiments are complete and results are convincing. Although more results/ analysis can be added, the paper is self-contained in the current stage.

---

> ### Author Response · Authors · 2021-11-18
> **Thank you for your detailed and constructive review!**
>
> Thank you for your detailed and constructive review! We appreciate your support for our paper, and for finding our formulation of the problem and insights for object detection task novel. Here are our brief replies to your questions:
> 1. About the design choice on the spatial featurizer and detector:
>     1. We indeed train the spatial featurizer and detector together in an end-to-end manner (Sec. 3.4).
>     2. About updating the offline history features: Thanks for bringing this up! In the current Hindsight design, if more data is observed during deployment, we will need to re-compute the SQuaSH features using the trained spatial featurizer with the latest T_max traversals. It should be a simple forward (inference) pass only. However, we think with careful design on the voxel-wise aggregation function, one can use exponential moving average to update the SQuaSH without re-read and re-compute the old data. This is an exploration direction that we are looking into for a future work.
> 2. About the process of getting dense point clouds: your understanding of the process is mostly correct - we assume a fixed global coordinate system and transform all point clouds to that coordinate system, thus they are naturally aligned. We will clarify this part in the final version. We plot visualization of the dense point clouds in Figure 1(a) under the “Past Traversals” Column.
> 3. Analysis of the pedestrians: Thanks for this suggestion! We tried to quantitatively evaluate the gain by looking at the precision-recall curve of different classes (Figure 2 and Figure 4), and we found that Hindsight generates significantly much less false positive detection, especially for the pedestrian class. This is also qualitatively verified in our visualizations. We do not observe the model has a significant difference on detecting moving/static pedestrains. We will look more into the detection results in the coming week and will update if we have new findings.

---

> > ### Comment · Reviewer_9mnc · 2021-12-06
> > **RE**
> >
> > Hi there, thanks for the replies, and sorry for late response!
> >
> > I am mostly positive about the paper except the few ablation or clarification that I mentioned in my review. One minor comment though: when more data are observed, except for using the new data to update the offline features with the trained model, is it possible to **finetune** the spatial featurizer with the additional data (maybe in a joint training scheme)?
> >
> > Thanks for taking time to address those issues in a revised version; the paper is already mostly complete at submission. More clarity and illustration as well as extra ablation is needed in a revision and I look forward to see the final publication.

---

### Decision · Program_Chairs · 2022-01-20

**Decision:**

Accept (Poster)

**Comment:**

The paper proposes a framework for object detection on lidar scans, with query of scene feature extracted offline from previous traversals. Overall there is good agreement among reviewers, with three recommending accepting the paper and one marginally accepting it -- to me the authors satisfactorily addressed most aspect raised in reviewing.